# Cooperative Spectrum Sensing with Coded and Uncoded Decision Fusion under Correlated Shadowed Fading Report Channels

**DOI:** 10.3390/s19010051

**Published:** 2018-12-23

**Authors:** Lucas dos Santos Costa, Dayan Adionel Guimarães, Rausley Adriano Amaral de Souza, Roberto César Dias Vilela Bomfin

**Affiliations:** 1National Institute of Telecommunications—Inatel Av. João de Camargo 510, Santa Rita do Sapucaí, MG 37540-000, Brazil; dayan@inatel.br (D.A.G.); rausley@inatel.br (R.A.A.d.S.); 2Vodafone Chair Mobile Communication Systems, Technische Universität Dresden, D-01062 Dresden, Germany; roberto.bomfin@ifn.et.tu-dresden.de

**Keywords:** cognitive radio, correlated shadowing, maximum eigenvalue detection, sensor information fusion, shadowed fading, spectrum sensing

## Abstract

This article addresses the impact of forward error correction when applied to the report channel transmissions of a centralized decision fusion cooperative spectrum sensing scheme designed to detect idle orthogonal frequency division multiple access (OFDMA) subchannels. The OFDMA signal is transmitted over slow frequency-selective multipath Rayleigh fading channels and sensed using the maximum eigenvalue detection test statistic. The decisions on the OFDMA subchannel occupancy are transmitted to a fusion center over report channels represented by a shadowed fading model combining a three-dimensional spatially correlated shadowing with a slow and flat multipath Rayleigh fading. Binary Bose-Chaudhuri-Hochquenghem (BCH) and Repetition codes are used to protect these decisions. Results show that shadowing correlation severely deteriorates the overall spectrum sensing performance and that error correction may not be able to protect the report channel transmissions. It can be even worse with respect to the system performance especially at low signal-to-noise regimes. In the situations in which error correction is effective, the Repetition code is capable of outperforming the BCH, meaning that the diversity gain may be more relevant than the coding gain when the spectrum sensing decisions are subjected to correlated shadowing.

## 1. Introduction

The scarcity of radio-frequency spectrum due to the unprecedented increased demand for new wireless communication systems and services has become a problem of paramount importance, especially in the case of wide-band communications. As a consequence, the creation of solutions and new standards in order to alleviate the spectrum occupation and to facilitate system interoperability is required. Moreover, the current fixed spectrum allocation policy, which only benefits licensed users, also called primary users (PUs), has been the major reason for the underutilized spectral resources, since studies have shown that the licensed bands are mostly vacant during certain periods of time and geographic locations [1].

The cognitive radio (CR) [2] concept has come to take advantage of the spectrum underutilization by allowing CR-enabled secondary user (SU) terminals to access unoccupied bands that are licensed to the PUs. The identification of unoccupied bands (also known as white spaces or spectral holes) is made by means of spectrum sensing [3], which subsequently enables dynamic opportunistic access by the SUs forming the secondary network. Thus, the efficiency of radio-frequency spectrum usage is ameliorated through the spectrum-sharing capabilities provided by this novel dynamic access policy.

Several spectrum sensing techniques have been proposed so far, which can be classified as narrowband and wideband according to the bandwidth of the spectrum sensed. Narrowband sensing techniques are limited to detect the presence of primary signals in a single narrow portion of the spectrum, whereas wideband techniques aim at jointly or sequentially monitoring multiple contiguous or non-contiguous portions of the spectrum. In what concerns narrowband sensing, energy detection, matched filter detection, and cyclostationary feature detection are widely discussed in the literature [3]. For wideband sensing, recent studies point to three major techniques: energy detection [4,5], wavelet-based detection [6], and compressed (or compressive) sensing detection [7].

An SU with spectrum sensing capability, in addition to possibly taking advantage of a spectrum occupation database, must be able to identify white spaces accurately to maximize the secondary network throughput and to avoid harmful interference to the primary network. The spectrum sensing process can be performed independently by each SU, but this approach suffers from reduced accuracy due to problems such as receiver uncertainty, multipath fading, and correlated shadowing [8]. Receiver uncertainty occurs when an SU cannot be reached by the PU signal, for instance, due to signal blockage, and this SU can erroneously decide that the sensed band is vacant. Multipath fading occurs in mobile communication systems due to the alternation between constructive and destructive interference among the received signals coming from multiple propagation paths. Shadowing is the local mean received signal power variations about the area mean PU signal powers received by the SUs, which is caused by natural or man-made obstacles in the signal path. Since multipath fading and shadowing may produce low received signal levels, they also contribute to erroneous decisions on the absence of the PU signal.

The SUs can cooperate to yield more accurate decisions on the occupation state of the sensed band. This cooperation is called collaborative spectrum sensing, or cooperative spectrum sensing (CSS). It can be centralized, distributed, or relay-assisted [8,9]. In centralized CSS with data fusion, data collected by each SU (e.g., samples from the received signal) are sent to a fusion center (FC) through dedicated control channels, usually referred to as report channels. The FC then processes the received data and makes a global decision upon the occupation state of the sensed band. In centralized CSS with decision fusion, only the decision on the channel occupation state made by each SU is transmitted to the FC, saving resources of the typically low-bandwidth report channels. The global decision is made by the FC by means of logic operations on all received SU decisions. Typical logic operations are AND, OR, and majority-voting (hereafter denoted by MAJ) [8,9]. In both centralized schemes, the global decision is informed back to the SUs, and the access algorithm adopted by the secondary network takes place.

The performance of the spectrum sensing is usually analyzed by means of the probability of detection and the probability of false alarm. The former is the probability that the sensed band is considered occupied when it is indeed occupied. The latter is the probability that the band is considered occupied when it is in fact vacant. The overall spectrum sensing performance is highly dependent on the conditions of the channels between the PU transmitter and the SUs, as well as on the conditions of the report channels between the SUs and the FC. For instance, in mobile wireless communications, which is the main stage for cognitive radio applications, multipath fading, and shadowing arise as important impairments that limit the spectrum sensing performance [10,11,12,13]. In addition, the actual spectrum sensing dynamics is influenced by SUs that might be moving, and at different distances from one another. This dynamics not only induces time-varying and spatially varying channel fading and shadowing but also produces different degrees of spatial correlation of the shadowing process. Thus, it is essential to assess the spectrum sensing performance when the system is subjected to such impairments. The three-dimensional (3D) spatially correlated shadowed fading channel model proposed in [14,15] is adopted in this article to encompass such impairments, and to yield a realistic assess of the spectrum sensing performance.

The majority of the third generation (3G) broadband wireless communication systems is based on direct sequence spread spectrum (DSSS), such as evolution data optimized (EVDO) and high speed packet access (HSPA). Fourth generation (4G) systems mostly adopt multicarrier transmission techniques, such as orthogonal frequency division multiplexing (OFDM), sometimes combined with its access counterpart, the orthogonal frequency division multiple access (OFDMA) [16]. OFDM is particularly attractive for delivering high speed data, especially over frequency-selective fading channels. Moreover, combined with the subcarrier nulling flexibility of OFDM signals, OFDM-based SUs can opportunistically access non-contiguous spectrum holes.

Several waveforms are being considered as potential candidates for the fifth generation (5G) air interfaces [17]. They include, but are not limited to, the discrete Fourier transform spread-OFDM (DFT-s-OFDM), which is already used in the 4G LTE uplink, the generalized DFT-spread-OFDM (G-DFT-s-OFDM) [18], the zero-tail DFT-spread-OFDM (zero-tail DFT-s-OFDM) [19], the unique-word DFT-spread-OFDM (UW DFT-s-OFDM) [20], the filter bank multi-carrier (FBMC) [17,21], and the generalized frequency division multiplexing (GFDM) [22].

From above, it can be noticed that the OFDM is prevalent in the 5G candidate waveforms and shall certainly remain as the root framework for new 5G waveform designs [23,24]. Thus, under the vision that cognitive radio networks will coexist with, or will be part of, 5G networks, it is of paramount importance that cognitive SU terminals are capable of sensing OFDM-like signals. The PU signals considered in the present article are OFDMA signals, a choice that is lined up with the present and forthcoming technologies.

The subsequent sections of this article are organized as follows: Section 2 explores the spectrum sensing framework, aiming at enlightening the problem tackled in the article. The related work and the contributions of the present research are given in Section 3. Section 4 is devoted to the local spectrum sensing problem, briefly describing the process of detecting idle OFDMA subchannels under the approach that is discussed at length in [25]. Section 5 is dedicated to the transmission of the local decisions to the fusion center, describing the report channel and the forward error correction (FEC) models proposed for assessing the performance of the centralized, decision-fusion cooperative spectrum sensing when subjected to errors in the report transmissions. Numerical results and interpretations are given in Section 6. Section 7 concludes the article and suggests some directions for future related research.

## 2. Spectrum Sensing Framework

Figure 1 provides an illustrative representation of the spectrum sensing topology considered in this article. The PU transmitter is part of the primary network, whereas the SUs and the FC belong to the secondary cognitive network. This figure is the reference for the rest of this section.

The OFDMA PU signal is transmitted over slow frequency-selective multipath Rayleigh fading sensing channels. When the PU transmitter is active, each SU receives a corrupted version of the OFDMA signal affected by the sensing channel fading and the additive white Gaussian noise (AWGN).

The PU signal is detected in each SU by means of the maximum eigenvalue detection (MED) test statistic [26], which is formed from the eigenvalues of the received signal sample covariance matrix. In each SU, the MED test statistic is compared with a local predefined decision threshold, yielding a local decision about the occupation state of each sensed OFDMA subchannel. The decision threshold is typically set according to a target constant false alarm rate (CFAR) of the local decisions.

Coded or uncoded SU decisions are sent to the FC through orthogonal report channels using binary phase-shift keying (BPSK) modulation. Perfect time and frequency synchronization are assumed. The FC makes coherent detection of the received signals and, for coded transmissions, it performs decoding of the Repetition or the Bose-Chaudhuri-Hochquenghem (BCH) encoded data. From the decoded data, the FC applies one of the three binary logic rules (AND, OR, or MAJ) to arrive at the final decision about the occupation states of the sensed OFDMA subchannels.

The whole process is depicted in Figure 2, where the spectrum sensing and report is illustrated for a single SU. The same process applies to the other SUs in cooperation.

The adoption of the spectrum sensing scenario described herein is justified in the remaining paragraphs of this section.

Frequency selectivity in the sensing channels is assumed due to the fact that an OFDM signal has a wide bandwidth, typically larger than the coherence bandwidth of the channel [27] (p. 221). Assuming that the spectrum sensing interval is smaller than the coherence time of the channel [27] (p. 222), the slow fading model is justified for the sensing and the report channels. The adoption of the Rayleigh distribution for the fading magnitude, which yields uniformly distributed phase variations, is justified by the fact that this type of fading represents a severe condition with no line-of-sight (LOS) between the PU transmitter and the SUs receivers [27] (p. 212).

It is considered that there is no shadowing affecting the sensing channels. The suppression of the shadowing from the sensing channel model is due to the following two reasons: (i) for simplicity, the main focus of this work is on the fusion of the spectrum sensing information; (ii) the influence of the spatially correlated shadowing on the performance of the local spectrum sensing is well explored in the literature; see, for instance, [28] and references therein.

The distance from the PU transmitter to the SUs is assumed to be much larger than the distances among the SUs, so that the distance-dependent propagation path losses from the PU transmitter to the SUs receivers can be considered approximately the same, yielding the same average snr at the input of each SU receiver.

The report channels, which are the main focus of the present work, are represented by a realistic channel model grounded on the shadowed fading approach proposed in [14]. This model, which is based on spatial grid points, capitalizes the well-known two-dimensional (2D) model discussed in [29], combining a three-dimensional spatially correlated shadowing with a slow flat Rayleigh fading. The report channel fading is considered flat because such control channels are typically low-rate and narrowband. This contrasts with the channels between the PU and the SUs, which convey the primary network wideband OFDMA signals.

The transmissions of the adjacent report bits sent from the SUs to the FC are affected by independent and identically distributed (i.i.d.) Rayleigh fading samples, since it is assumed that the report data are subjected to a sufficiently long time-interleaving. This interleaving is feasible since the report frame structure has many more bits than the report data itself [30,31]. The interleaving breaks the burst errors caused by the fading channel memory effect, which in turn allows for the effective operation of the FEC codes; recall that FEC codes reduce their error correction capability if bit errors are grouped in bursts, which may exceed the maximum number of correctable errors in a codeword or block [27] (Section 8.2).

Binary BCH and Repetition codes are used to protect the SU decisions. The Repetition code is a trivial code [32], i.e., it produces no coding gain over a pure AWGN channel, but it is very simple and does produce diversity gain over fading channels [33,34]. Both codes have been investigated to allow for a broader analysis concerning the role of the diversity gain and the coding gain in the global spectrum sensing performance. Moreover, the BCH encoding and decoding process is a matured technology that is being used for decades in real communication systems [35]. In fact, the pursuit of efficient decoding algorithms for BCH codes is still and active research topic [36].

Instead of using BCH codes, one could argue why modern error-correcting codes, such as Turbo [37], low density parity check (LDPC) [38], and Polar codes [39], were not considered in the present context. These are powerful codes usually referred to as capacity-achieving codes, which can allow the system operation very close to the Shannon limit. However, this outstanding performance is achieved at the cost of using very long codewords, typically having tens of thousands bits. The cooperative spectrum sensing application prevents the use of such long codes for three main reasons: (i) the sensing interval must be as short as possible in order to maximize the secondary network throughput and to confer agility to the spectrum sensing task (but not too short to yield poor spectrum sensing performance); (ii) the short sensing interval is divided into the sensing phase itself and in the report phase, which in turn is even shorter; (iii) the transmission rate over a report channel is typically low. The combination of these reasons leads to the conclusion that the number of bits that can be accommodated within the report interval is small. Hence, the spectrum sensing application demands the use of short codes. A BCH code with short codewords is a proper candidate, since it can outperform Turbo, LDPC, Polar, and other powerful codes with similar short lengths [40].

According to [8], when binary local decisions are reported to the FC, applying a linear decision fusion rule to obtain the global decision is convenient. The Chair–Varshney rule is considered the optimal one [41], but it demands the knowledge of prior probabilities associated to the primary transmitter activity. The AND, the OR, and the majority-voting (MAJ) suboptimal decision fusion rules are adopted in the present work, since they are the most common and the most simple ones, yet do not demand the knowledge of the PU transmitter activity. These rules are special cases of the general *N*-out-of-*M* rule, in which the FC decides in favor of the presence of the PU signal in the sensed band if, among the *M* SUs, *N* or more decides in favor of an active PU transmitter. For N=M, the *N*-out-of-*M* rule becomes the AND rule, for N=1 it becomes the OR rule, and for N=⌊M/2+1⌋ it becomes the MAJ rule, ⌊x⌋ being the largest integer smaller than or equal to *x*.

After the global decision upon the state of each OFDMA subchannel is reached, the SUs are informed back by the FC about this state. Subsequently, an appropriate access mechanism is carried out by the secondary network if a given subchannel is declared vacant. The channel access and other posterior tasks are not considered in this article; the analysis is restricted to the overall spectrum sensing process alone.

## 3. Related Work and Contributions

Only a few references can be found in the literature when dealing with the design or the application of FEC schemes to protect the transmissions from the SUs to the FC. For instance, in [42], a joint channel decoding and decision fusion strategy is applied to the report channel transmissions, combining a Neyman–Pearson-based test statistic with an extended Hamming code. Similarly to [42], the joint channel decoding and decision fusion strategy is also applied in [43], but using LDPC codes instead of extended Hamming codes. The results of these references show that the spectrum sensing performance can be improved when compared with the case in which channel decoding and decision fusion are separately made.

A larger number of references can be found in other contexts, for example in sensor network applications. To mention a few of them, the BCH code is used in [44] to protect video signal transmissions from the sensor nodes to a central node. In [45], correlated quantities gathered by the sensor nodes are transmitted to a common destination via AWGN multiple access channels using a concatenation of a BCH with a low density generator matrix (LDGM) code, with the BCH acting to reduce the error floor produced by the LDGM.

In [34,46], a memoryless binary symmetric channel (BSC) has been adopted to simulate report channel errors in a centralized CSS scheme designed to detect OFDMA signals. The eigenvalue fusion proposed in [25] is confronted with decision fusion schemes under such errors. The eigenvalues are digitized before transmission to the FC, while the SU decisions are Repetition-coded for error protection and then sent to the FC. A trade-off analysis on the report channel data traffic and the spectrum sensing performance is then established, showing that SU decisions are more sensitive to the report channel errors, but the amount of redundancy added by the channel coding process does not always lead to a larger report channel data traffic in comparison with the eigenvalue fusion. Thus, a case-by-case analysis is needed to decide upon which of these two fusion schemes should be adopted.

A comparative analysis between the performances of hard decision and soft decision approaches for CSS in the presence of report channel errors is given in [47]. The optimal fusion rule for the soft decision is derived, and the distribution of the corresponding test statistic is established. As a complement of [47], the article [48] gives a detailed analysis of the bit error probability walls for the logic decision fusion rules under the assumption of i.i.d. report channel errors.

It is worth emphasizing that [34,46,47,48,49,50,51,52] implicitly or explicitly adopt a simple discrete-time memoryless BSC model for assessing the influence of report channel errors in the global spectrum sensing performance. However, the error statistics in real report channels may depart significantly from those produced by a memoryless BSC model. The main causes of discrepancy are the channel memory effect and the signal fluctuations caused by the combination of multipath fading and spatially correlated shadowing. As a consequence, errors may occur in bursts in deep fading circumstances, and with high probabilities when deep shadowing occurs, impacting the global decisions at the FC. Moreover, it is well-known that the performance of FEC codes are deeply affected by the error distribution [53]. Hence, to analyze the influence of report channel errors and to develop proper countermeasures, it is of major importance to adopt a model capable of closely representing a real channel. One out of three main approaches can be used in this representation: (i) a discrete-time binary symmetric channel model with memory, (ii) a continuous-time waveform channel model, or (iii) a continuous-time-based vector channel model. The accuracy of the first approach depends on how close the model represents the overall influence of multipath fading and correlated shadowing, which can lead to a very intricate modeling process. The second approach is simpler, but it is normally more demanding in terms of computational burden in simulations. The third is intermediate in terms of both computational burden and complexity, and for this reason it is adopted in the present work.

This article addresses the performance of a centralized decision fusion CSS scheme designed for detecting idle OFDMA subchannels in SU networks. The main contributions are as follows:The performance analysis is carried out considering a realistic model to simulate the report channel behavior. None of the previously described works have adopted such a model; the prevalent choice was to consider a memoryless binary symmetric channel.Since the report channel impairments affect the global decision made at the FC due to errors produced in the decisions that are transmitted from the SUs to the FC, FEC codes are applied as an attempt to protect the SU decisions. The short codes selected reflect the time-limited nature of the report events. Only a few references have addressed, so far, the problem of protecting the report channel transmissions under the combined effect of fading and spatially correlated shadowing.From the analyses and results presented in the article, important conclusions are drawn regarding the overall spectrum sensing performance and the use of FEC codes in the report transmissions, highlighting the main differences between these transmissions and those made in conventional digital communication systems. To the best of the authors knowledge, no such analyses have been made so far.

## 4. Detection of Idle OFDMA Subchannels

The OFDMA is a multiple access technique that allocates a set or multiple sets of subcarriers to a given user, allowing for the simultaneous access by several users to the allocated service band. One set of frequencies is referred to as one subchannel. Basically, a subchannel can be formed according to two methods: the adjacent subcarrier method (ASM), which groups contiguous subcarriers to form a subchannel, and the diversity subcarrier method (DSM), in which non-contiguous subcarriers are grouped to form a subchannel.

When any spectrum sensing scheme is applied to the detection of a primary OFDMA signal, it typically aims at detecting the signal at the subchannel level, i.e., it aims at detecting if a given subchannel is vacant or not.

Let an OFDMA signal formed by $K^{\prime }$ subcarriers and *S* subchannels. A number $K=K^{\prime }/S$ subcarriers will form a subchannel indexed by *s*, s=1,2,⋯,S. The spectrum sensing of an OFDMA time-domain signal performed in each SU at the *s*-th subchannel level can be formulated as a binary hypothesis test represented by
(1)$$y_{m}\left(t\right)=\{\;\begin{array}{cc} w_{m}\left(t\right) & :H_{0,s}\\ h_{m}\left(t\right)*x\left(t\right)+w_{m}\left(t\right) & :H_{1,s} \end{array},$$
where $y_{m}\left(t\right)$ is the continuous-time received signal at the *m*-th SU, $h_{m}\left(t\right)$ is the impulse response of the channel between the primary transmitter and the *m*-th SU, the symbol `*’ represents the convolution operation, x(t) is the continuous-time primary transmitted signal, and $w_{m}\left(t\right)$ is the continuous-time zero-mean AWGN at the *m*-th SU receiver input. The hypotheses $H_{0,s}$ and $H_{1,s}$ denote the absence and the presence of the PU signal in the *s*-th subchannel, respectively.

The global probability of detection and the global probability of false alarm at the FC, associated to the *s*-th OFDMA subchannel, are respectively $P_{\mathrm{dFC},s}=\mathrm{Pr}\left\{\mathrm{decision}=H_{1,s}|H_{1,s}\right\}$ and $P_{\mathrm{faFC},s}=\mathrm{Pr}\left\{\mathrm{decision}=H_{1,s}|H_{0,s}\right\}$. The former is the probability that the *s*-th subchannel is declared occupied when, indeed, the primary transmitter is active in that subchannel. The latter is the probability that the *s*-th subchannel is declared occupied when, actually, it is vacant.

It is assumed that each of the *M* SUs in cooperation knows the subcarrier allocation map for each subchannel, an information that can be obtained from the primary network standard specifications. Thus, all SUs know which portions of the spectrum must be sensed.

A matrix of order K×J with frequency-domain samples from the *s*-th subchannel is formed at the *m*-th SU according to
(2)$$A_{s}^{\left(m\right)}=\left[\begin{array}{ccc} Y_{1,1}^{\left(m\right)}\left(s\right) & \ldots & Y_{1,J}^{\left(m\right)}\left(s\right)\\ \vdots & \ddots & \vdots \\ Y_{K,1}^{\left(m\right)}\left(s\right) & \cdots & Y_{K,J}^{\left(m\right)}\left(s\right) \end{array}\right],$$
where *J* is the number of samples collected by each SU in each subcarrier, $Y_{k,j}^{\left(m\right)}\left(s\right)$ denotes the *j*-th sample collected by the *m*-th SU in the *k*-th subcarrier of the *s*-th subchannel, for j=1,2,⋯,J, m=1,2,⋯,M, and k=1,2,⋯,K. The next step is to compute *S* sample covariance matrices of order K×K, according to
(3)$$R_{s}^{\left(m\right)}=\frac{1}{J}A_{s}^{\left(m\right)}A{{}_{s}^{\left(m\right)}}^{†},$$
where † denotes complex conjugate and transpose.

From the *s*-th covariance matrix, *K* eigenvalues are estimated by the *m*-th SU and ordered as $\left\{\lambda_{1,m,s}\geq \lambda_{2,m,s}\geq \cdots \geq \lambda_{K,m,s}\right\}$. The occupation of each subchannel is determined in each SU by comparing the predefined decision threshold γ with the MED test statistic, which is [25]
(4)$$\begin{array}{c} T_{\mathrm{MED},s}^{\left(m\right)}=\frac{\lambda_{1,m,s}}{\sigma^{2}} \end{array},$$
$\sigma^{2}$ being the additive white Gaussian noise variance. If $T_{\mathrm{MED},s}^{\left(m\right)}>\gamma$, the *s*-th subchannel is declared occupied by the *m*-th SU; if $T_{\mathrm{MED},s}^{\left(m\right)}<\gamma$, the *m*-th SU declares the *s*-th subchannel vacant.

The *S* subchannels can be sensed by each SU in a single sensing interval as long as the whole OFDMA signal is sampled in the frequency domain, and each sample covariance matrix defined in Equation (3) is formed in parallel to the other ones, by simply selecting the appropriate frequency bins associated to the corresponding subchannel.

The resulting MS local decisions are sent to the FC, where each group of *M* decisions are logically combined by means of the rules AND, OR, or MAJ, yielding the global decision upon the occupancy of each of the *S* subchannels.

## 5. Coded Report Channel Transmissions

This section deals with the transmission of the local SU decisions to the FC, describing the channel and the FEC models proposed for assessing the performance of the spectrum sensing under transmission errors.

### 5.1. The Role of the Error Control on the Global Spectrum Sensing Performance

For any FEC scheme, the relation between the average SNR per coded bit and the average SNR per uncoded bit is $E_{c}/N_{0}=rE_{b}/N_{0}$, where *r* is the code rate, which is the ratio between the message block length at the input of the encoder and the codeword length at its output, $E_{c}$ is the average energy per coded bit, in joules, $E_{b}$ is the average energy per input message bit, in joules, and $N_{0}=2\sigma^{2}$ is the AWGN power spectral density, in Watts per hertz. In decibels, $E_{c}/N_{0}=10\log_{10}\left(r\right)+10\log_{10}\left(E_{b}/N_{0}\right)$ dB. Thus, channel coding reduces the bit energy from the encoder input to the encoder output, increasing the probability of a coded bit error. However, the error correction capability of nontrivial codes can produce a net reduction in the message bit error probability after decoding (hereafter denoted as the decoded bit error probability), compensating for the increase of the coded bit error probability. In the case of trivial codes, this compensation does not exist, meaning that there is no coding gain.

Typically, an error-correcting code exhibits a crossing point in the decoded bit error probability at a given average SNR per bit: below this SNR, the coded error probability is not compensated for by the error correction capability of the code, and the decoded bit error probability becomes larger than the one without channel coding; above this SNR, the coding gain starts to show up, yielding a decoded bit error probability smaller than the one without channel coding. The value of the SNR in which this crossing point occurs varies according to the type of code and to its decoding algorithm, as will be exemplified later on in this paper.

Hereafter assuming that the local spectrum sensing performances achieved by the SUs are the same and are equal for all OFDMA subchannels, the subscript *s* can be dropped from the notation. Thus, let $P_{\mathrm{dSU}}$ and $P_{\mathrm{faSU}}$ respectively denote the probability of detection and the probability of false alarm achieved by each SU for any sensed OFDMA subchannel. These probabilities are governed by the test statistic adopted in each SU, which in this article is the MED defined in Equation (4). When report channel errors occur, some SU decisions received at the FC are modified, as if the probabilities $P_{\mathrm{dSU}}$ and $P_{\mathrm{faSU}}$ were different. Denote the corresponding equivalent probabilities seen by the FC, taking into account the report channel errors, as $P_{\mathrm{dSU}}^{\prime }$ and $P_{\mathrm{faSU}}^{\prime }$. The equivalent and the actual probabilities of detection and false alarm are related through [5]
(5)$$\begin{array}{cc} P_{\mathrm{dSU}}^{\prime } & =P_{\mathrm{dSU}}\left(1-P_{b}\right)+P_{b}\left(1-P_{\mathrm{dSU}}\right), \end{array}$$
(6)$$\begin{array}{cc} P_{\mathrm{faSU}}^{\prime } & =P_{\mathrm{faSU}}\left(1-P_{b}\right)+P_{b}\left(1-P_{\mathrm{faSU}}\right), \end{array}$$
where $P_{b}$ is the bit error probability after decoding at the FC, which depends on the report channel characteristics, on the performance of the adopted error correcting code, and on the modulation. For instance, if an uncoded transmission with BPSK modulation is made over a slow (i.e., approximately constant during the modulation symbol) and flat (i.e., no frequency selective) Rayleigh fading channel, $P_{b}$ is given by [27] (Equation (6.200)), which is
(7)$$P_{b}=\frac{1}{2}\left(1-\sqrt{\frac{\Gamma }{1+\Gamma }}\right),$$
where $\Gamma =E\left[\alpha^{2}\right]E_{b}/N_{0}$ is the average SNR per bit, with $E[\alpha^{2}]$ being the second moment (average channel power gain) of the Rayleigh fading envelope α.

If the local decisions about the occupation state of each OFDMA subchannel are i.i.d., the global probabilities of detection and false alarm at the FC under the *N*-out-of-*M* rule with hard decisions are, respectively [5],
(8)PdFC=∑ℓ=NMMℓ(PdSU′)ℓ(1−PdSU′)M−ℓ,
(9)PfaFC=∑ℓ=NMMℓ(PfaSU′)ℓ(1−PfaSU′)M−ℓ.

The role of the error control coding applied to the report channel transmissions is to reduce $P_{b}$ in Equations (5) and (6), improving the global spectrum sensing performance metrics given in Equations (8) and (9).

Sometimes it is more convenient to assess the effect of some system parameter variation on the joint variation of the probabilities of detection and false alarm, which is captured by the global probability of decision error, given by
(10)$$P_{\mathrm{error}}=P_{\mathrm{faFC}}P_{H_{0}}+\left(1-P_{\mathrm{dFC}}\right)P_{H_{1}},$$
where $P_{H_{0}}$ and $P_{H_{1}}$ are the probabilities associated with the inactivity and activity states of the primary transmitter, respectively.

### 5.2. Block-Coded Report Transmissions

Let $n_{B}$ and $n_{R}$ be the codeword lengths of the bch and the Repetition code, respectively, and let $\kappa_{B}$ and $\kappa_{R}$ be the lengths of the input message block carrying the SU decisions applied to the BCH and the Repetition encoders, respectively. The bits that represent the set of decisions made by each SU on the occupation state of the *S* OFDMA subchannels are encoded by means of $\left(n_{R},\kappa_{R}\right)=\left(n_{R},1\right)$ Repetition codes with configurable coding rate $r_{R}=1/n_{R}$, odd $n_{R}$, or by ($n_{B},\kappa_{B}$) binary bch codes with $\kappa_{B}=S$.

The coded decisions are mapped into baseband BPSK symbols according to the rule: bit 1 $\Rightarrow +\sqrt{E_{c}}=+\sqrt{rE_{b}}$, bit 0 $\Rightarrow -\sqrt{E_{c}}=-\sqrt{rE_{b}}$, where $E_{b}$ and $E_{c}$ are the average energies per uncoded and coded bit, respectively, and $r=r_{B}$ or $r=r_{R}$, depending on the block code used.

### 5.3. Spatially Correlated Shadowing Model

The 3D spatially correlated shadowed channel model originally proposed in [14] is briefly explained in this subsection. In order to analyze the influence of the report channel errors on the performance of the spectrum sensing, here this model replaces the simple binary memoryless report channel model adopted in the related works described in Section 3, which are [34,46,47,48,49,50,51,52].

The correlated shadowing model is depicted in Figure 3. It is based on grid points that form a 3d space having L×L×L cubic meters. These grid points are references that establish locations with null pairwise spatial correlation. In other words, if two SUs are located at two grid points, they will be subjected to totally uncorrelated shadowing. For this reason, the distance between adjacent grid points is referred to as the decorrelation distance [29], denoted by $D_{\mathrm{dec}}$. In practice, $D_{\mathrm{dec}}$ is on the order of tens of meters [54]. When $D_{\mathrm{dec}}$ is increased, any pair of SUs separated by a given distance greater than zero becomes subjected to more correlated shadowing. The opposite occurs when $D_{\mathrm{dec}}$ is decreased.

One of the cubes that form the entire space on the left in Figure 3 is depicted on the right. The position of the *m*-th SU is established by means of the coordinates $x_{m}$, $y_{m}$, and $z_{m}$, measured with respect to a reference grid point of each cube, which is the one marked with *A* in the single cube. If $z_{m}=0$, this 3D model specializes to the traditional 2D one considered in [29]. The positions of ${\mathrm{SU}}_{1}$ and ${\mathrm{SU}}_{2}$ are shown as an example. In this example, the spatial correlation between the fading processes affecting ${\mathrm{SU}}_{1}$ and ${\mathrm{SU}}_{2}$ will be nonzero, since they are influenced by the same set of grid points.

A zero-mean Gaussian random variable is associated to each grid point, forming a set of ${\left(L/D_{\mathrm{dec}}+1\right)}^{3}$ iid variables. For example, the random variables *A*, *B*, *C*, *D*, *E*, *F*, *G*, and *H* are associated to the grid points located on the vertexes of the single cube on the right in Figure 3, which is a zoomed version of one of the cubes on the left. The standard deviations of the ${\left(L/D_{\mathrm{dec}}+1\right)}^{3}$ Gaussian random variables are $\sigma_{\mathrm{dB}}$, which is the standard deviation of the lognormal shadowing.

The spatial correlation of the shadowing process that affects the *m*-th SU is produced by a linear combination of the eight Gaussian random variables associated to the grid points that surrounds this SU. For instance, if *A* to *H* are such random variables, a shadowing sample is generated as
(11)$$S_{m}=\left[\left(A{\tilde{x}}_{m}^{\prime }+B{\tilde{x}}_{m}\right){\tilde{y}}_{m}^{\prime }+\left(C{\tilde{x}}_{m}^{\prime }+D{\tilde{x}}_{m}\right){\tilde{y}}_{m}\right]{\tilde{z}}_{m}^{\prime }+\left[\left(E{\tilde{x}}_{m}^{\prime }+F{\tilde{x}}_{m}\right){\tilde{y}}_{m}^{\prime }+\left(G{\tilde{x}}_{m}^{\prime }+H{\tilde{x}}_{m}\right){\tilde{y}}_{m}\right]{\tilde{z}}_{m},$$
where ${\tilde{x}}_{m}=x_{m}/D_{\mathrm{dec}}$, ${\tilde{y}}_{m}=y_{m}/D_{\mathrm{dec}}$, and ${\tilde{z}}_{m}=z_{m}/D_{\mathrm{dec}}$ are normalized distances to the reference grid point, and ${\tilde{x}}_{m}^{\prime }=1-{\tilde{x}}_{m}$, ${\tilde{y}}_{m}^{\prime }=1-{\tilde{y}}_{m}$, and ${\tilde{z}}_{m}^{\prime }=1-{\tilde{z}}_{m}$.

It can be noticed that the weights in the linear combination given by Equation (11) are distance-dependent and normalized, i.e., $0\leq {\tilde{x}}_{m},{\tilde{y}}_{m},{\tilde{z}}_{m}\leq 1$. It means that if two SUs are close to each other, even if they are located in different neighboring cubes, their shadowing will be composed by a combination of some common Gaussian random variables (due to common grid points), generating statistical correlation. For instance, this closeness reproduces the situation in which two SUs are behind the same obstacle that blocks their signals to the FC.

In order to guarantee that the standard deviation of $S_{m}$ is equal to the desired value of shadowing standard deviation $\sigma_{\mathrm{dB}}$, it is necessary to make the correction [55]
(12)$$S_{m}^{\prime }=\frac{S_{m}}{\sqrt{\left(1\;-\;2{\tilde{x}}_{m}\;+\;2{{\tilde{x}}_{m}}^{2}\right)\left(1\;-\;2{\tilde{y}}_{m}\;+\;2{{\tilde{y}}_{m}}^{2}\right)\left(1\;-\;2{\tilde{z}}_{m}\;+\;2{{\tilde{z}}_{m}}^{2}\right)}},$$
which finally produces the Gaussian random variable that represents the shadowing channel between the FC and the *m*-th SU.

In summary, the model just described is capable of generating correlated Gaussian samples whose correlation depends on the three-dimensional SUs coordinates and on the value of $D_{\mathrm{dec}}$.

### 5.4. Shadowed Fading Report Channel Model

The report channels considered in this article applies the shadowed fading model of [56], combining the spatially correlated lognormal shadowing described in the previous subsection with a small-scale multipath Rayleigh fading. In discrete-time representation, the report channel gain matrix [15,56] is defined by
(13)$$H=\frac{\mathrm{abs}\left(H^{f}+H^{s}\right)}{\sqrt{2b_{0}+\exp\left[\frac{2\sigma_{\mathrm{dB}}^{2}}{{\left(20\log_{10}e\right)}^{2}}\right]}},$$
where the operator abs(·) denotes the absolute value of the elements in the underlying matrix, $H^{f}\in C^{M\times n}$ and $H^{s}=h^{s}1^{T}\in C^{M\times n}$, in which 1 is the n×1 vector of ones, and the superscript T denotes transposition, with *n* being the transmitted block size. This block size is n=S if no channel coding is employed; if the bch code is adopted, $n=n_{B}$; when the Repetition code is used, $n=n_{R}S$. The matrices $H^{f}$ and $H^{s}$ represent the multipath fading and the shadowing, respectively. The elements of $H^{f}$ are iid complex Gaussian random variables, i.e., the fading magnitude obeys a Rayleigh distribution. The second moment of the fading magnitude is $2b_{0}$. The vector $h^{s}$ that defines $H^{s}$ is formed by the report channel shadowing samples $h_{m}^{s}$, in which $\arg\left\{h_{m}^{s}\right\}∼U\left(0,2\pi \right]$, and $|h_{m}^{s}|=10^{S_{m}^{\prime }/20}$, with $S_{m}^{\prime }$ being the shadowing random variables with standard deviation $\sigma_{\mathrm{dB}}$ obtained from Equation (12). In this equation, the normalized coordinates ${\tilde{x}}_{m}$, ${\tilde{y}}_{m}$, and ${\tilde{z}}_{m}$ of the *m*-th SU are determined according to the spatial distribution of the SUs in the 3D space illustrated in Figure 3.

The samples in $H^{s}$ affect all bits transmitted from each SU to the FC during a report round, i.e., the shadowing process is slow enough to be considered constant during the report interval. Contrasting with this matrix, the fading matrix $H^{f}$ is formed by i.i.d. components, i.e., each reported bit, coded or not, is subjected to i.i.d. multipath fading. This is owed to the assumption of perfectly time-interleaved decision bits to prevent burst errors, allowing for the proper effectiveness of the FEC codes. Without FEC, this time-interleaving approach does not produce any effect on the average bit error probability. Hence, the Rayleigh fading is said to be slow and flat with respect to each reported bit, coded or not.

Finally, it is worth highlighting that, in the formation of the report channel matrix H according to Equation (13), the absolute value operation in the numerator is applied to model a coherent detection of the received BPSK symbols at the FC, i.e., the phase rotation that would be produced by the channel is removed. The argument of the square root in the denominator of Equation (13) is the second moment of the combined channel fading and shadowing, meaning that the role of this denominator is to guarantee unitary average channel power gain, averaged over the fading and the shadowing variations. This normalization allows for an easy configuration of the average SNR per decoded bit at the FC, denoted by $E_{b}/N_{0}$, simply setting the noise power accordingly.

### 5.5. Received Signal Model and Decoding

Assume that the *S* decisions upon to the occupation state of the OFDMA subchannels, as determined by each of the *M* SUs, are arranged in a matrix of order M×S. The *S* decisions from each SU, when encoded by the BCH code with rate $r_{B}=S/n_{B}$, yield the matrix $C\in R^{M\times n_{B}}$ of coded SU decisions, whose elements are $\pm \sqrt{r_{B}E_{b}}$. When the Repetition code with rate $r_{R}=1/n_{R}$ is used, each SU decision is encoded into $n_{R}$ bits, resulting in a total of $n_{R}S$ coded bits, thus yielding a matrix $C\in R^{M\times n_{R}S}$ of coded SU decisions, whose elements are $\pm \sqrt{r_{R}E_{b}}$.

The corrupted coded symbols received at the FC are represented by the matrix B, whose order is the same of C, computed according to
(14)B=H∘C+V,
where H is the channel matrix defined in Equation (13), V is the matrix of additive Gaussian noise samples whose order is the same of B and C, and the symbol ∘ denotes the Hadamard product (element-wise multiplication).

The BPSK detector at the FC outputs real-valued samples (soft information) associated to the received coded symbols. Each codeword-size block containing these samples is then applied to the BCH or the Repetition decoder, which implements brute force (exhaustive search) maximum likelihood (ML) soft-decision decoding, without channel state information. It can be shown that, in the case of the Repetition code, this ML decoding is equivalent to the equal gain combining (EGC) of the soft information associated to the received codeword symbols. In the case of the BCH, the ML decoding is chosen because its performance can be closely approximated in practice by well-known and well-established algorithms, such as ordered statistics decoding (OSD) [35,57]. Other recent soft-decision decoding algorithms of BCH codes also closely approximate the ML decoding performance, as the one considered in [36].

The decoded bits, or the uncoded and corrupted SUs decisions in the case of no channel coding, are subsequently combined according to the desired decision fusion rule (AND, OR, or MAJ voting) to yield the global decision about the occupation of each OFDMA subchannel.

## 6. Numerical Results and Discussion

The spectrum sensing scenario considered in this section is composed of a primary network with S=5,6, and 7 OFDMA subchannels, and $K^{\prime }=25,30$, and 35 subcarriers, respectively. Thus, the number of subcarriers per subchannel is constant and given by $K=K^{\prime }/S=5$. The subcarriers per subchannel are randomly selected, meaning that the DSM allocation mode is adopted. The number of SUs in cooperation is M=6. The PU signal power is considered unitary, and the SNRs of the received PU signal at the SUs are set to −10 dB. This small SNR regime is chosen to represent a more degrading but yet realistic situation from the perspective of the spectrum sensing performance [30].

The sensing channel between the PU transmitter and each SU receiver is modeled as a 25, 30, or 35-path slow frequency-selective Rayleigh fading channel whose frequency response is kept constant during a sensing interval, being varied independently from one sensing round to the next. The second moment of the sensing channel gains is normalized to 1 to keep the average received signal power equal to the average transmitted signal power. The number of samples collected by each SU in each subcarrier frequency is J=120.

A spectrum sensing performance result is usually displayed by means of a receiver operating characteristic (ROC) curve, which trades the probability of false alarm and the probability of detection, and sometimes by means of the area under curve (AUC), which gives the area under a ROC curve, or yet by means of the decision error probability defined in Equation (10). Here, each value on a roc curve was obtained from 50,000 Monte Carlo simulation events carried out using the Matlab^®^ software (version R2018a). Each event corresponds to the following steps:send an OFDM PU signal through *M* independent fading channels to the SUs, under a PU transmitter activity of 50% ($P_{H_{0}}=P_{H_{1}}=0.5$);perform, in each SU, the local spectrum sensing over all OFDMA subchannels by means of the MED test statistic computed from J=120 received samples collected in each subcarrier, and for a predefined local decision threshold γ;report the coded or uncoded SUs decisions to the FC via orthogonal spatially correlated shadowed fading channels;perform channel decoding and make the global decision on a single OFDMA subchannel;compute false alarm and detection rates, which are the estimates of the associated probabilities.

The above steps were repeated by varying γ, so that the ROC curves were traced out. In the fourth step, the occupancy of a single OFDMA subchannel has been monitored, since it suffices to compute the false alarm and detection rates. This can be done owing to the i.i.d. channel effects across the subchannels, to the i.i.d. test statistics on the subchannels, and, hence, to the independence and equal distributions of the corresponding decisions.

Aiming at investigating the correlated shadowed fading effect over the report channel transmissions, two scenarios for system simulation were explored: the first one assumes no channel coding; the second considers the use of the BCH or the Repetition code. In both scenarios, the standard deviation of the shadowing, $\sigma_{\mathrm{dB}}$, is 6.14, 8.68, or 12.28 dB, which can be translated into weak, weak-to-moderate, and severe shadowing, respectively. These standard deviations are in the range of outdoor [58] and indoor [59] measurements. The parameter $b_{0}$ in Equation (13) is set to 0.21. These values were chosen following [60]. The length *L* of the sides of the 3D space depicted in Figure 3 is equal to 50 m. The *M* SUs in cooperation are uniformly distributed within this space. The decorrelation distance $D_{\mathrm{dec}}$ is 10, 50, and 90 m to simulate small, medium, or high shadowing correlation; the scenario of totally uncorrelated shadowing is included in the present analysis as well.

### 6.1. Results without Channel Coding

Figure 4 shows global spectrum sensing performances without applying channel coding to the report channel transmissions, under severe shadowing. The influence of the spatially correlated shadowing is also analyzed by varying the decorrelation distance. The error-free and the pure-Rayleigh ROC curves are also given to allow for a clear perception of performance degradations caused by the small-scale fading alone and by the large-scale shadowing combined with the small-scale fading.

For the MAJ, OR, and AND rules, the average SNR per decoded bit is $E_{b}/N_{0}=6$ dB, which corresponds to an uncoded bit error probability $P_{b}\approx 0.053$ in a pure-Rayleigh channel, as computed by Equation (7) with $E[\alpha^{2}]=1$, i.e., $\Gamma =E_{b}/N_{0}$.

As a first observation regarding Figure 4, it can be seen that $P_{\mathrm{faFC}}$ and $P_{\mathrm{dFC}}$ are lower or upper bounded in some situations, which is in agreement with the theoretical results in [9,51]. For instance, taking into account the or rule, $P_{\mathrm{faFC}}\geq 1-{\left(1-P_{b}\right)}^{M}$ and this bound does not depend on the snr [9]. A careful observation of Figure 4b confirms that $P_{\mathrm{faFC}}$ is around 0.2787 if the OR rule is considered over the pure-Rayleigh channel with $P_{b}\approx 0.053$, which is consistent with [9].

As far as the decorrelation distance $D_{\mathrm{dec}}$ is concerned, it can be noticed in Figure 4a that the global spectrum sensing performances under the MAJ rule are close to each other for $D_{\mathrm{dec}}=10$, 50, and 90 m, but a larger $D_{\mathrm{dec}}$ (larger spatial correlation) produces a greater performance penalty, as expected. However, the opposite behavior occurs when the OR or the AND rules are applied, as depicted in Figure 4b,c, respectively. Interestingly enough, the increase in the shadowing spatial correlation actually improves the spectrum sensing performance, although the net performance degradation due to shadowing is by far more severe than in the case of the MAJ rule.

The phenomenon of performance improvement with increased shadowing spatial correlation can be justified in light of [61], where the *N*-out-of-*M* rule is addressed in the context of systems reliability. As stated in this reference, the effect of correlation on reliability can be beneficial as well as detrimental. The model for the global spectrum sensing decision based on the *N*-out-of-*M* rule is analogous to the model for reliability analysis under this rule, which means that the same concept applies, i.e., the effect of the shadowing correlation on the global decision can be beneficial as well as detrimental. A correlation-robust rule is also devised in [61]. It states that, if *N* is chosen as the integer closest to p(M−1)+1, the *N*-out-of-*M* rule becomes almost immune to correlation, where *p* is the probability of a decision in favor of $H_{1}$, that is $p=P_{\mathrm{faFC}}P_{H_{0}}+P_{\mathrm{dFC}}P_{H_{1}}$. For instance, since here were adopted $P_{H_{0}}=P_{H_{1}}=0.5$ and M=6, if $P_{\mathrm{faFC}}=0.1$ and $P_{\mathrm{dFC}}=0.9$, it follows that N=3 or N=4 will make the decisions almost insensitive to correlation. Notice that N=4 is the one adopted here to configure the MAJ rule. In this case, Figure 4a verifies the robustness of the MAJ rule to the variation of $D_{\mathrm{dec}}$, and its superior performance when compared to the OR and AND rules is shown in Figure 4b,c.

The derivation of the probability distribution of the number of successes in the sequence of correlated binary trials that represent the SU decisions as seen by the FC remains an open problem. This is a formidable challenge, not only from the perspective of the generalization of the binomial distribution to correlated binary trials, as discussed in [61], but also from the difficulty in mapping the correlation coefficient onto the shadowed fading report channel characteristics. As a complement to this challenge, the derivation of the error probability of the *N*-out-of-*M* rule in this scenario is also of major importance; a related analysis is presented in [47].

Table 1 shows the AUCs related to Figure 4 and to other simulation results not depicted in Figure 4, considering weak, weak-to-moderate, and severe correlated shadowing with $D_{\mathrm{dec}}=10$, 50, and 90 m (Table Parts I, II, and III, respectively). The AUC is adopted as the performance metric to make the presentation more concise, since analyses via ROC curves would be considerably difficult due to the large amount of graphs. An AUC=1 is related to a perfect system performance, for which the ROC knee is located at $P_{\mathrm{dFC}}=1$ and $P_{\mathrm{faFC}}=0$. For $P_{\mathrm{dFC}}=P_{\mathrm{faFC}}$, the roc is a diagonal line between the points (0,0) and (1,1), called the line of no discrimination or random guess line. The corresponding AUC is 0.5.

For spectrum sensing, it is known that a ROC curve lying on the random guess line means that the system is useless. Although an AUC=0.5 may suggest a performance on the random guess line, one can observe that some AUC values in Table 1 are below 0.5, which could be erroneously translated in an unacceptable system performance. However, in all occurrences of an AUC below 0.5, the associated ROC curve laid above the random guess line. For example, even for the worst case depicted in Figure 4c for $D_{\mathrm{dec}}=10$, which refers to an AUC=0.018959 in Table 1, the ROC curve lies above the random guess line, indicating that some operating point on this ROC is usable, i.e., it is better than a random guess.

In order to emphasize the performance degradation caused by the shadowing, let us have a look at the AUCs associated to Figure 4a–c, which are {0.918633,0.681236,0.472566} for the pure-Rayleigh curves and {0.402746,0.127475,0.018959} for the worst case of the respective fusion rules. It can be noticed that, due to the shadowing, the AUCs were reduced in 56.16%, 81.29%, and 95.99%, respectively. During deep shadowing, the received $E_{b}/N_{0}$ is lowered, increasing the local mean bit error probability, thus causing a performance penalty to the global spectrum sensing performance.

Similar comparisons can be easily made in light of Table 1, considering other situations in terms of the shadowing standard deviation and the decorrelation distance. The corresponding comments were omitted here, for the sake of conciseness.

### 6.2. Results with Channel Coding

Figure 5 shows ROC curves for coded report transmissions under severe correlated shadowing, i.e., $\sigma_{\mathrm{dB}}=12.28$, as well as under the pure-Rayleigh channel, for the three decision fusion rules in analysis with $E_{b}/N_{0}=6$ dB. In order to allow for a fair comparative analysis, the average redundancy per message bit, n/κ, was chosen to be as close as possible for both codes in each rule. Specifically, for the MAJ rule in Figure 5a, $n_{B}/\kappa_{B}=n_{R}/\kappa_{R}=3$; for the AND rule in Figure 5c, $n_{B}/\kappa_{B}=n_{R}/\kappa_{R}=9$; and for the OR rule in Figure 5b, $n_{B}/\kappa_{B}\approx 5.17$ and $n_{R}/\kappa_{R}=5$. The adopted BCH codes that can achieve the above redundancies per message bit are (15, 5), (31, 6), and (63, 7), respectively. The Repetition codes are respectively (3, 1), (5, 1), (9, 1). The codes (15, 5) and (3, 1) are less powerful and have been used with the MAJ rule because this is the best rule among the three under analysis. The performance of the OR rule typically lies between the performances of the MAJ and the AND rule, and this is the reason for using the intermediate pair of codes (31, 6) and (5, 1). The codes (63, 7) and (9, 1) are the most powerful and have been applied to the AND rule. Figure 6 illustrates the performances of the above-mentioned codes over the full-interleaved flat Rayleigh fading channel, with maximum-likelihood soft-decision detection, without channel state information. This figure unveils the expected superior performance of the BCH over the Repetition code, as well as exemplifies the crossing point beyond which the error correction starts to be effective: it occurs around 0 dB for the BCH codes and far below −5 dB (hence not shown) for the Repetition codes.

Comparing the curves in Figure 5 for the pure-Rayleigh report channel in Figure 4 and Figure 5, it can be seen that the BCH code was able to correct all bit errors introduced by the channel. In this case, the BCH outperformed the Repetition code, which can be credited to a dominance of the coding gain of the BCH over the diversity gain of the Repetition code when the report channel is free of shadowing. From the curves associated to the presence of shadowing, one can infer that the use of FEC codes to protect the SU decisions seems to be useless for a report channel under this impairment. In fact, lower performance levels with respect to the uncoded transmission results shown in Figure 4 are observed. Nonetheless, even in this unfavorable situation, the superiority of the Repetition code over the BCH is evident.

Similarly to what has been done in Table 1 concerning the uncoded transmissions, Table 2 presents results in terms of AUCs considering coded transmissions with the BCH and the Repetition code, for $D_{\mathrm{dec}}=10$, 50, and 90 m, L=50 m, and $\sigma_{\mathrm{dB}}=6.14$, 6.68, and 12.28 dB.

Comparing the AUC values in Table 2 with those in Table 1, for the BCH and Repetition codes, it can be seen that the majority of them unveil no performance gain of the BCH coded transmissions over the uncoded ones. These values are in the gray-shaded areas of Table 2. Even for the smallest shadowing standard deviation, $\sigma_{\mathrm{dB}}=6.14$ dB, the BCH code failed with its purpose for the AND rule with $D_{\mathrm{dec}}=10$ and $D_{\mathrm{dec}}=50$ m and for the maj rule with $D_{\mathrm{dec}}=90$ m. For the Repetition code, however, it can be clearly noticed that the FEC scheme was effective in general. This can now be credited to the dominance of the diversity gain of the Repetition code over the coding gain of the BCH code when the report channel is under shadowing.

For the AUC values outside of the gray-shaded areas in Table 2, the AUCs are larger than the respective ones in Table 1, meaning that, under weak shadowing, the use of FEC schemes seems to be mildly effective in the setting of $E_{b}/N_{0}=6$ dB. However, a larger AUC cannot always be translated into a better performance in terms of ROC when, for instance, a target probability of false alarm or detection, or both, is established. For example, when the Repetition code is applied under $\sigma_{\mathrm{dB}}=12.28$ dB, with $D_{\mathrm{dec}}=90$ for the MAJ rules, and $D_{\mathrm{dec}}=10$ for the OR and the AND rules, the AUCs are those boldfaced, i.e., 0.405737, 0.127083, and 0.019908, respectively. Using $P_{\mathrm{faFC}}=0.1$ as a reference, in Figure 5, it can be seen that $P_{\mathrm{dFC}}\approx 0.36$ with the MAJ rule; neither this reference $P_{\mathrm{faFC}}$ nor the corresponding $P_{\mathrm{dFC}}$ can be attained with the OR rule, and $P_{\mathrm{dFC}}\approx 0.15$ with the AND rule. Similarly, for a reference value of $P_{\mathrm{dFC}}=0.65$, it follows that $P_{\mathrm{faFC}}\approx 0.2$ with the MAJ rule, and this reference $P_{\mathrm{dFC}}$ and the corresponding $P_{\mathrm{faFC}}$ cannot be attained with the OR or the AND rules. Only the MAJ rule with the Repetition code could be able to satisfy a system requirement of, for example, $P_{\mathrm{faFC}}\leq 0.1$ and $P_{\mathrm{dFC}}\geq 0.3$.

Given the results herein, it can be concluded that the use of an FEC code as an attempt to protect the transmissions of the local SUs decisions to the FC might not bring considerable improvements to the global spectrum sensing performance. This behavior can be credited to the typical SNR crossing point, below which the coded bit error probability is not compensated for by the error correction capability of the code, meaning that the decoded bit error probability becomes larger than the one without channel coding; see Section 5.1 and Figure 6. Indeed, the SNR above which the decision fusion process becomes almost insensitive to the report channel errors is relatively low when compared to the one that is necessary to yield typical error rates demanded in conventional wireless digital communication systems. Hence, to be more effective, the FEC code must exhibit such a crossing point in very low SNRs, which is a characteristic of Turbo, LDPC, and Polar codes. Unfortunately, these codes typically have very long codewords and time-consuming encoding or decoding processes, being incompatible with the short time interval constraints imposed by the decision fusion of spectrum sensing information.

To close this section, Figure 7 shows global probabilities of decision errors estimated via Monte Carlo simulations in the cases of coded report transmissions over the pure-Rayleigh and the shadowed fading report channels, and in the cases of coded and uncoded transmissions over the shadowed fading channel, as a function of the average SNR per bit at the FC receiver, $E_{b}/N_{0}$, for the decision fusion rules MAJ (a), OR (b), and AND (c). The theoretical probabilities of decision error over the error-free report channels and for the uncoded transmissions over the pure-Rayleigh channel were obtained via Equations (5)–(10). The FEC codes considered in Figure 7 are the same adopted to plot Figure 5. The spatially correlated shadowed fading was assumed to be in a severe situation, with $\sigma_{\mathrm{dB}}=12.28$ dB and with $D_{\mathrm{dec}}=10$ m.

The SNR of the sensing channel, which governs the local spectrum sensing performance and, as a consequence, also governs the global performance, was adjusted so that $P_{\mathrm{error}}=0.05$ in the error-free report channel for each decision fusion rule. This value was computed using the pair of $P_{\mathrm{dFC}}$ and $P_{\mathrm{faFC}}$ returning the smallest $P_{\mathrm{error}}$ for each of the fusion rules. In other words, the operating point on a global ROC curve was chosen as the one yielding the best global spectrum sensing performance in the error-free report channel condition, which corresponds roughly to the knee of the curve and which is approximately the point closest to the ideal values of $P_{\mathrm{faFC}}=0$ and $P_{\mathrm{dFC}}=1$. Hence, the variation of $E_{b}/N_{0}$ moves the knee of the ROC from the worst $P_{\mathrm{error}}=0.5$ (for $P_{\mathrm{faFC}}=P_{\mathrm{dFC}}=0.5$), toward the smallest $P_{\mathrm{error}}$ whose value is attained in the error-free report channel.

The reference decision error probability $P_{\mathrm{error}}=0.05$ in the error-free report channel condition is half of the $P_{\mathrm{error}}$ attained in the case of the standardized targets $P_{\mathrm{faFC}}=0.1$ and $P_{\mathrm{dFC}}=0.9$ [30], assuming $P_{H_{0}}=P_{H_{1}}=0.5$, according to Equation (10).

Figure 7 broadens the interpretations obtained from Figure 4 and Figure 5, and provides a stronger practical value to the numerical results because the spectrum sensing performance is now assessed over a wide range of $E_{b}/N_{0}$ values. Firstly, notice in Figure 7 that $P_{\mathrm{error}}$ is predominantly smaller when the report channel is the pure-Rayleigh, for all fusion rules in analysis, as expected. It can also be seen that the FEC schemes produced performance gains only for some (higher) values of $E_{b}/N_{0}$; more expressive gains are observed in the case of the pure-Rayleigh report channel, and at lower values of $E_{b}/N_{0}$ than in the case of the shadowed fading.

In the case of the MAJ rule, as depicted in Figure 7a, it can be observed that neither the BCH nor the Repetition code produced significant performance gains in the shadowed fading channel scenario; a small gain can be noticed in the case of the pure-Rayleigh channel, around $E_{b}/N_{0}=5$ dB. It must be emphasized that this small gain is particular to the MAJ rule; it has not been produced due to the worst pair of codes used in this case: the BCH (15, 5) and the Repetition (3, 1), a choice that has been made due to the superior performance of the MAJ rule with respect to the AND and OR, as shown in Figure 4 and Figure 5. This behavior is due to the fact that the report channel bit error probability that is sufficient for the MAJ rule to achieve a target $P_{\mathrm{error}}$ is considerably larger than for the OR and the AND rules, a situation in which the coding and the diversity gains are smaller (refer to Figure 6). This means that, for instance, if the BCH (63, 7) and the Repetition (9, 1) codes were used with the MAJ rule, a considerably small performance improvement would be obtained beyond that already produced by the BCH (15, 5) and the Repetition (3, 1) .

On the other hand, it can be noticed in Figure 7b,c that the codes applied to the OR and AND rules, respectively the BCH (31, 6) with the Repetition (5, 1) and the BCH (63, 7) with the Repetition (9, 1), yielded considerable performance improvements. This is credited to the fact that, to reach a target $P_{\mathrm{error}}$, the OR and the AND rules need to operate under smaller bit error probabilities than the MAJ rule, where coding and diversity gains are more pronounced. In other terms, the channel coding is more effective when applied to the decision fusion rules that, to a greater extent, require such coding to be effective.

As also expected, the Repetition code is not capable of producing the same performance improvements achieved with the BCH code. However, the Repetition code is effective (i.e., the attained $P_{\mathrm{error}}$ is smaller with an error control than without it) over practically the entire range of $E_{b}/N_{0}$ values, which does not occur in the case of the BCH code. This happens because the crossing point where the Repetition code starts to produce diversity gain (refer to Section 5.1 and Figure 6) occurs at much lower $E_{b}/N_{0}$ values than the crossing point where the BCH starts to produce coding gain. Thus, in the values of $E_{b}/N_{0}$, where the BCH is ineffective, a large performance degradation with respect to the uncoded report transmissions is produced for all fusion rules. Owing to the fact that any code exhibits a performance crossing point somewhere in the $E_{b}/N_{0}$ scale, channel coding will be useless and even deleterious to system performance if this point occurs at an $E_{b}/N_{0}$ value already sufficient for the target global spectrum sensing performance to be attained. This is more likely to occur in the case of the MAJ decision fusion rule.

Keeping in mind that an error correcting code applied to report channel transmissions must be short, another important aspect must be equally considered: the decoding complexity. Recall that maximum likelihood soft-decision decoding without channel state information was applied, which is a very simple process in the case of the Repetition code but carries considerable complexity in the case of the BCH [35]. One could think of using hard-decision decoding to reduce complexity, but this would penalize the performance considerably. The performance penalty produces a shift of the above-mentioned crossing point to the right, eventually making the code useless in the $E_{b}/N_{0}$ range of interest. Thus, it can be concluded that the soft-decision decoded Repetition code is an attractive solution, since it aggregates the small complexity to a global spectrum sensing performance that, in the $E_{b}/N_{0}$ range of interest, might not be too far from the one achieved with a soft-decision decoding of a more powerful code with comparable length.

## 7. Conclusions and Opportunities for Further Research

This article addresses the performance of a centralized decision fusion CSS scheme. The coded or uncoded SU decisions were transmitted to the FC over report channels represented by a realistic shadowed fading model that combines a three-dimensional spatially correlated shadowing with a slow and flat multipath Rayleigh fading. Binary BCH and Repetition error correcting codes were used as an attempt to protect the transmitted decisions. The decision fusion rules AND, OR, and majority-voting (MAJ) were analyzed. Results unveiled that, as expected, shadowing correlation degrades the reliability of the global decisions made at the FC. Although only two FEC schemes have been tested, which prevents a general conclusion about their impacts, the extensive simulation results demonstrated that the use of an FEC code in the decision fusion process might be not capable of protecting the SU decisions. Indeed, the use of an FEC scheme can be even worse than not applying any FEC at all, especially at low report channel SNR regimes. The results also unveiled that, especially in the case of the OR and the AND rules, the Repetition code can outperform the BCH, meaning that the diversity gain may be more relevant than the coding gain when the SU decisions are subjected to correlated shadowing with full-interleaved fading. Large variations in the global spectrum sensing performance under different report channel parameters were also observed, emphasizing the importance of adopting channel models that consider the combined effect of additive noise, multipath fading, and spatially correlated shadowing. Among the decision fusion rules investigated, the best performance was achieved by the MAJ, no matter if the report channel is error-free or is under pure-Rayleigh or fading plus shadowing.

The conclusions drawn herein are further supported by the fact that the decision fusion process in the context of spectrum sensing significantly differs from a conventional data transmission process. In the former, high channel error rates are supported, especially in the case of the MAJ rule, in comparison with conventional data transmission because the aim is to make a global decision on the occupation state of the sensed channel, not to retrieve transmitted data. Thus, indeed, the effectiveness of channel coding schemes is expected to be quite different when applied to these processes. Moreover, the signal-to-noise ratios in which a satisfactory spectrum sensing performance can be attained is considerably smaller than those in which satisfactorily small data error rates can be achieved in data transmission systems; notice that the report channel signal-to-noise ratio might be smaller than the information theoretic limit below which the channel errors cannot be controlled.

The following opportunities for related research can be highlighted: (i) the derivation of the probability distribution of the number of successes in the sequence of correlated binary trials that represent the SU decisions, as seen by the FC, taking into account the shadowed fading channel statistics; (ii) the development of a discrete-time shadowed fading channel model according to this probability distribution and to the corresponding channel statistics; (iii) the derivation of the error probability of the *N*-out-of-*M* rule in the present scenario; (iv) the study of the impact of channel coding on the report channel transmissions in light of information theory; (v) the development of general channel code design rules for the specific fusion of hard spectrum sensing decisions and for digitized soft decisions, taking into account the need for short codes.

## Figures and Tables

**Figure 1 sensors-19-00051-f001:**
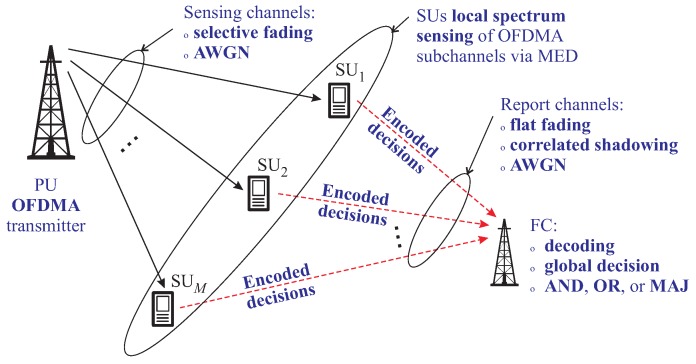
Centralized decision fusion CSS of OFDMA subchannels.

**Figure 2 sensors-19-00051-f002:**
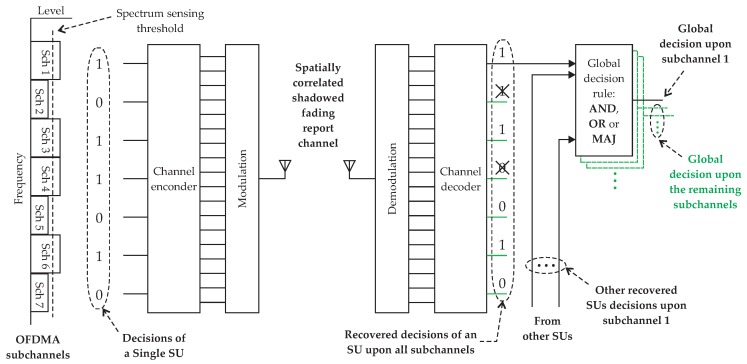
Didactic representation of the coded decision fusion process. The spectrum sensing and report is illustrated for a single SU.

**Figure 3 sensors-19-00051-f003:**
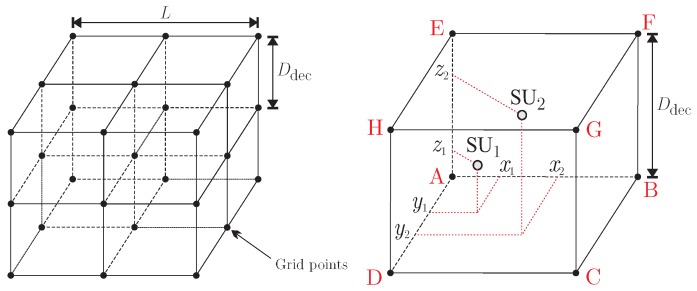
Pictorial representation of the three-dimensional correlated shadowing model: 3D space with L×L×L m${}^{3}$ (**left**); zoomed single cube with $D_{\mathrm{dec}}\times D_{\mathrm{dec}}\times D_{\mathrm{dec}}$ m${}^{3}$, with arbitrarily placed ${\mathrm{SU}}_{1}$ and ${\mathrm{SU}}_{2}$ (**right**).

**Figure 4 sensors-19-00051-f004:**
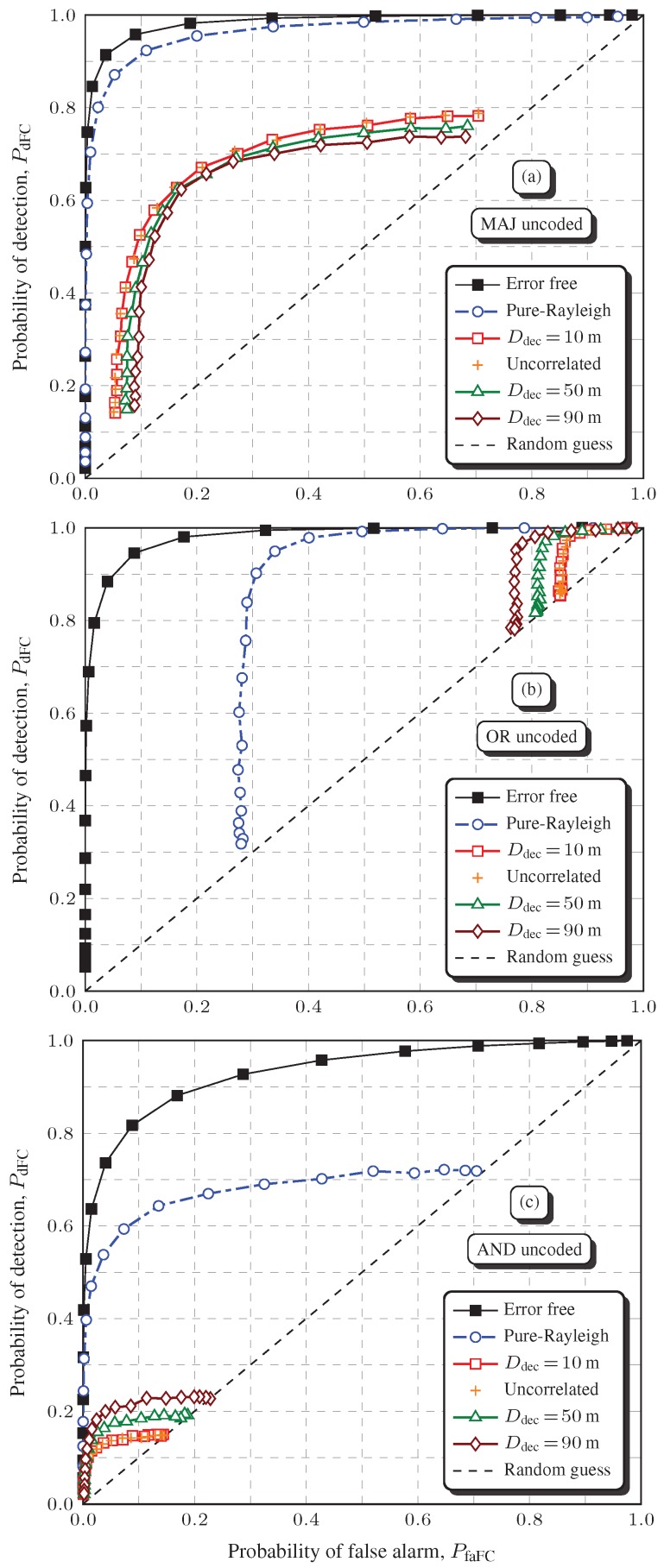
Rocs considering no channel coding applied to the report channel transmissions, for $E_{b}/N_{0}=6$ dB, $\sigma_{\mathrm{dB}}=12.28$ dB and L=50 m: MAJ rule (**a**), OR rule (**b**), and AND rule (**c**). This figure is better viewed in color.

**Figure 5 sensors-19-00051-f005:**
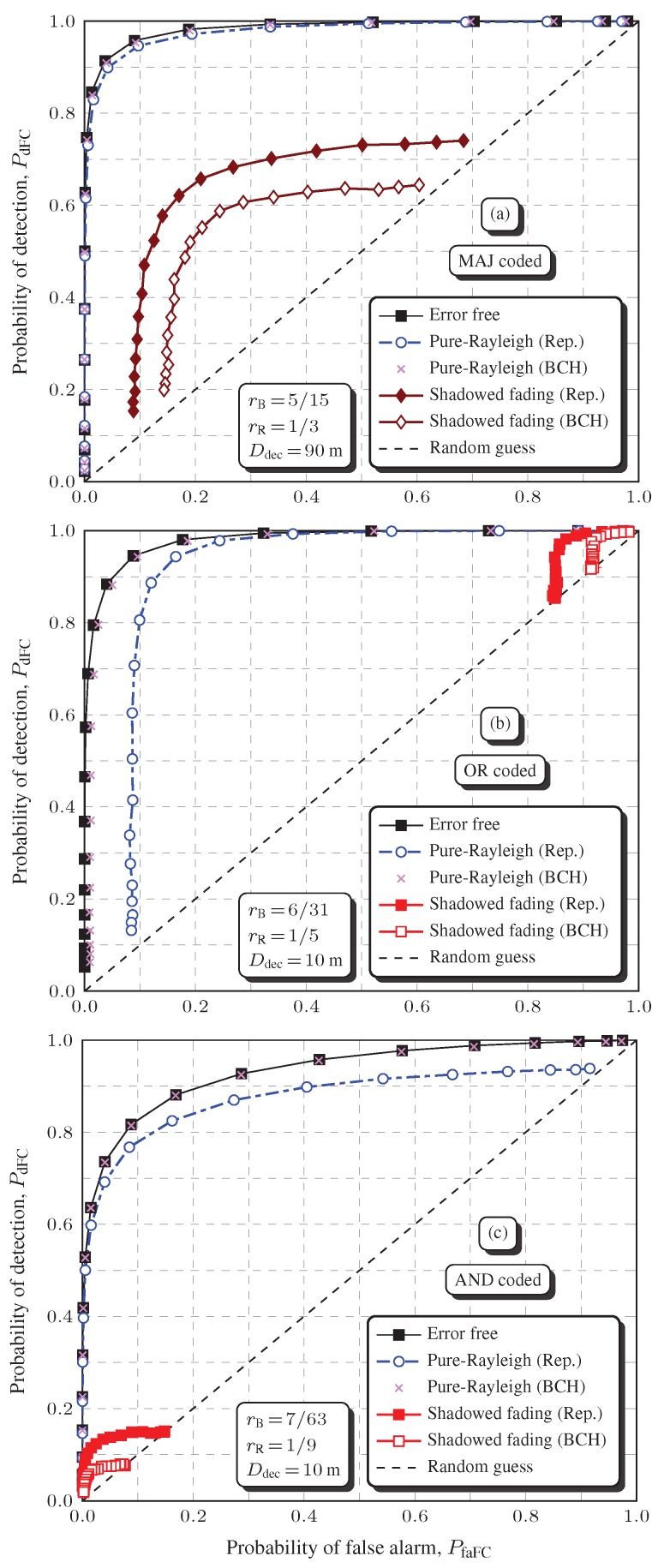
ROCs considering coded report channel transmissions via BCH and Repetition codes, for $E_{b}/N_{0}=6$ dB, $\sigma_{\mathrm{dB}}=12.28$ dB, and L=50 m: MAJ rule (**a**), OR rule (**b**), and AND rule (**c**). This figure is better viewed in color.

**Figure 6 sensors-19-00051-f006:**
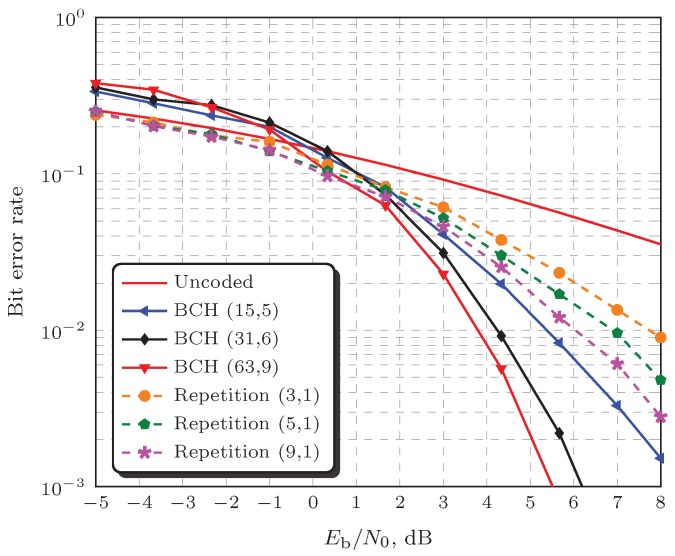
Performances of BCH and Repetition codes over the full-interleaved flat Rayleigh fading channel, with maximum-likelihood soft-decision detection without channel state information.

**Figure 7 sensors-19-00051-f007:**
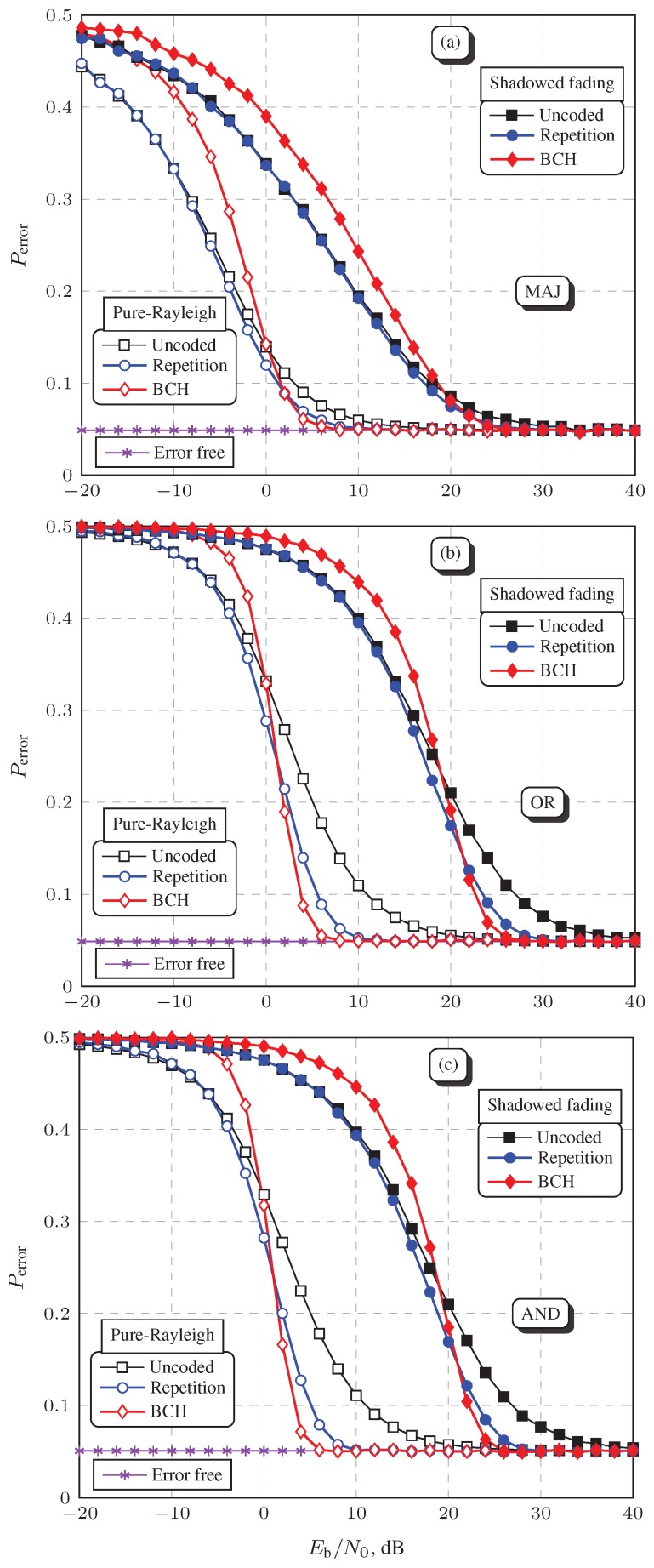
FC decision error rates under pure-Rayleigh and severe correlated shadowed fading, with uncoded and coded SUs transmissions via BCH and Repetition codes, for the MAJ rule (**a**), the OR rule (**b**), and the AND rule (**c**), all with $D_{\mathrm{dec}}=10$ m.

**Table 1 sensors-19-00051-t001:** AUCs for the uncoded MAJ, OR, and AND decision fusion rules, for $\sigma_{\mathrm{dB}}=6.14$, 8.68, and 12.28 dB, L=50 m, and $D_{\mathrm{dec}}=10$, 50, and 90 m.

**I.** $D_{\mathrm{dec}}=10$ **m**			
$\sigma_{\mathrm{dB}}$	**MAJ**	**OR**	**AND**
6.14	0.883682	0.578683	0.340961
8.68	0.772699	0.371024	0.146844
12.28	0.456074	0.127475	0.018959
**II. $D_{\mathrm{dec}}=50$ m**			
$\sigma_{\mathrm{dB}}$	**MAJ**	**OR**	**AND**
6.14	0.873795	0.593581	0.364219
8.68	0.748177	0.409234	0.176610
12.28	0.420448	0.171063	0.032338
**III. $D_{\mathrm{dec}}=90$ m**			
$\sigma_{\mathrm{dB}}$	**MAJ**	**OR**	**AND**
6.14	0.867578	0.607996	0.379690
8.68	0.728518	0.439383	0.201059
12.28	0.402746	0.206694	0.047992

**Table 2 sensors-19-00051-t002:** AUCs for the coded MAJ, OR, and AND decision rules, with $r_{B}=\left\{5/15,6/31,7/63\right\}$ and $r_{R}=\left\{1/3,1/5,1/9\right\}$, for $\sigma_{\mathrm{dB}}=6.14,8.68$, and 12.28 dB, L=50 m, and $D_{\mathrm{dec}}=10,50$, and 90 m. The gray-shaded areas and bold-faced values are explained in the text.

**I.** $D_{\mathrm{dec}}=10$ **m**			
**Repetition Code**			
$\sigma_{\mathrm{dB}}$	**MAJ**	**OR**	**AND**
6.14	0.904719	0.655512	0.445769
8.68	0.790084	0.402396	0.170776
12.28	0.451550	0.127083	0.019908
**BCH Code**			
$\sigma_{\mathrm{dB}}$	**MAJ**	**OR**	**AND**
6.14	0.895237	0.619083	0.409900
8.68	0.703429	0.276161	0.077101
12.28	0.324772	0.062240	0.005220
**II. $D_{\mathrm{dec}}=50$ m**			
**Repetition code**			
$\sigma_{\mathrm{dB}}$	**MAJ**	**OR**	**AND**
6.14	0.894063	0.669020	0.466375
8.68	0.765731	0.436177	0.202683
12.28	0.428467	0.171599	0.031920
**BCH Code**			
$\sigma_{\mathrm{dB}}$	**MAJ**	**OR**	**AND**
6.14	0.879542	0.650800	0.443212
8.68	0.659388	0.333784	0.117005
12.28	0.298985	0.105427	0.012644
**III. $D_{\mathrm{dec}}=90$ m**			
**Repetition Code**			
$\sigma_{\mathrm{dB}}$	**MAJ**	**OR**	**AND**
6.14	0.889968	0.681048	0.485556
8.68	0.748057	0.469948	0.226962
12.28	0.405737	0.208837	0.048480
**BCH Code**			
$\sigma_{\mathrm{dB}}$	**MAJ**	**OR**	**AND**
6.14	0.864465	0.675177	0.471522
8.68	0.629363	0.399938	0.160739
12.28	0.274431	0.161950	0.026191

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
