# Peer review of "Cooperative Spectrum Sensing with Coded and Uncoded Decision Fusion under Correlated Shadowed Fading Report Channels"

_sensors, 2018, doi:10.3390/s19010051_

Round 1

Reviewer 1 Report

The paper address the performance of a centralized decision fusion Cooperative spectrum sensing scheme. The coded or uncoded CR decisions are transmitted to the FC over report channels. Binary BCH and repetition error correcting codes were used as an attempt to protect the transmitted decisions.

The paper deals with interesting topic, and is reasonable presented. Comments/remarks include:

 Authors confuse the terminology regarding to cognitive radio, typically abbreviated as CR, and secondary users, typically abbreviated as SUs. In the text, authors use the abbreviation CR for both cases that makes the text sometimes difficult to understand.

Authors should clarify how the given work differs to previous authors works, what is new.

To improve the text comprehensibility, it would be useful to add an illustrative figure (a part of figure 1) that would explain the investigated issues and steps.

The considered code, BCH, seems to be a bit out of date. A more up-to-date codes (e.g., LDPC codes) should be considered in the investigations.

Does the analysis and simulation consider static or moving users (secondary users)?

Author Response

Please, see attached file.

Reviewer 2 Report

I cannot recommend the publication of the paper in its present form for the following reasons:

- The writing needs a careful revision.

- The adopted coding schemes are somewhat old-fashioned (BCH codes) or have poor performance (repetition codes). The authors should consider more powerful codes such as turbo codes or LDPC codes.

- The flat fading channel is not a good model for OFDM (or OFDMA) schemes. A more realistic channel model should be considered.

- Why using BPSK?

- The performance results are more-or-less the expected ones.

Author Response

Please, see attached file.

Round 2

Reviewer 1 Report

Authors took into account the reviewers' remarks and comments and the paper has been properly extended and inconsistencies improved. Thus, I recommend accepting the paper in the given form. 

Reviewer 2 Report

The authors addressed all my concerns.